# Clinical Features and Immunophenotypes of Double-Hit Diffuse Large B-Cell Lymphoma

**DOI:** 10.3390/diagnostics12051106

**Published:** 2022-04-28

**Authors:** Cheng-Han Wu, Jyh-Pyng Gau, Chieh-Lin Jerry Teng, Yu-Hsuan Shih, Yu-Chen Su, Ren-Ching Wang, Tsung-Chih Chen

**Affiliations:** 1Division of Hematology/Medical Oncology, Department of Medicine, Taichung Veterans General Hospital, Taichung 40705, Taiwan; wuchen50218@hotmail.com (C.-H.W.); drteng@vghtc.gov.tw (C.-L.J.T.); rollingstone07@gmail.com (Y.-H.S.); jennysunm@hotmail.com (Y.-C.S.); 2Division of Hematology and Oncology, Department of Medicine, Taipei Veterans General Hospital, Taipei 11217, Taiwan; jpgau@vghtpe.gov.tw; 3School of Medicine, National Yang Ming Chiao Tung University, Taipei 30010, Taiwan; 4Department of Life Science, Tunghai University, Taichung 40704, Taiwan; 5School of Medicine, Chung Shan Medical University, Taichung 40201, Taiwan; 6College of Medicine, National Chung Hsing University, Taichung 40227, Taiwan; 7Department of Pathology, Taichung Veterans General Hospital, Taichung 40705, Taiwan; renchingw@gmail.com; 8Department of Nursing, College of Nursing, Hungkuang University, Taichung 43302, Taiwan; 9Graduate Institute of Clinical Medicine, College of Medicine, National Taiwan University, Taipei 10617, Taiwan

**Keywords:** DLBCL, double-hit, immunophenotyped, prognosis, bulky disease

## Abstract

Double-hit (DH) genetics induces a reduction in the complete remission (CR) and, consequently, in poor overall survival (OS) in diffuse large B-cell lymphoma (DLBCL) patients. Unfortunately, DH identification is time-consuming. Here, we retrospectively reviewed 92 newly diagnosed DLBCL patients, stratified them into the DH (*n* = 14) and non-DH groups (*n* = 78), and compared their clinical features and outcomes. The results revealed that the DH group had a higher percentage of bulky disease than the non-DH group (64.3% vs. 28.2%; *p* = 0.013). More patients in the DH group tested positive for double expresser (DE) (50.0% vs. 21.8%; *p* = 0.044). The three-year OS rates of patients with and without DH were 33.3% and 52.2%, respectively (*p* = 0.016). Importantly, advance stage and multiple comorbidities were correlated with a high mortality rate in multivariate analysis. Furthermore, by combining DE and the bulky disease, a specificity of 89.7% for DH prediction was achieved. In summary, DH genetics, not DE immunopositivity, could be a factor for an inferior OS in DLBCL. A combination of bulky disease and a positive DE immunophenotype could facilitate DH genetics prediction in newly diagnosed DLBCL patients.

## 1. Introduction

Diffuse large B-cell lymphoma (DLBCL) is the most common subtype of non-Hodgkin lymphoma (NHL) in adults, accounting for approximately half of all new cases [1]. Immunochemotherapy with rituximab plus cyclophosphamide, doxorubicin, vincristine, and prednisone (R-CHOP) approximately cures two-thirds of patients with DLBCL [2,3]. However, in patients with refractory/relapsed lymphoma subjected to R-CHOP treatment, the outcomes remain dismal [3,4]. DLBCL is a heterogeneous disease. The morphology, cell of origin (COO), molecular features, and genetic profiles can differ among different DLBCL types [5]. Although R-CHOP is the standard of care for newly diagnosed DLCBL, intensified chemotherapeutic regimens could improve the outcomes of DLBCL with high-risk features [6,7,8]. Optimal risk stratification according to the nature of the disease could facilitate the development of an appropriate therapeutic strategy for DLBCL patients by balancing the efficacy and treatment-related toxicities. 

For decades, the predictive model of DLBCL using the Revised International Prognostic Index (R-IPI) score or Lugano classification has been based on clinical features, and the disease has extended. The purpose of these models is to predict the outcomes of DLBCL patients undergoing R-CHOP treatment by identifying patients with inferior overall survival (OS) [9]. Lugano classification incorporates disease extent and tumor burden to the NHL stage [10]. Regardless of the disease stage, the regimen containing immunochemotherapy is the standard treatment for DLBCL. However, 30–40% of patients eventually relapse after standard treatment and are refractory to immunochemotherapy [11]. Personalized treatment remains the best strategy for DLBCL.

With the improvement in genetic technologies, DLBCL has been discovered to be genetically heterogeneous. MYC rearrangement is one of the most important genetic alterations that have a significant impact on treatment outcomes [12,13]. Consequently, the World Health Organization 2017 Classification defined high-grade B-cell lymphoma with MYC and BCL2 and/or BCL6 rearrangements among the tumors within DLBCL morphology as double hit (DH) DLBCL [14]. In addition to DH, double expresser (DE) DLBCL, defined as MYC/BCL2 protein co-expression by immunohistochemical (IHC) staining, also has an aggressive clinical course in some reports [15]. However, the impact of DE may not be significant after adjusting for LDH levels [16]. Therefore, DH is a more profound prognostic factor for DLBCL than DE. A study by Riedell et al. [17] showed that DH DLBCL has a lower complete response (CR) rate than the non-DH DLBCL. The inferior CR rate leads to reduced OS.

Some studies have demonstrated that intensified chemotherapy or frontline autologous stem-cell transplantation may improve the outcome of DH DLBCL. However, the optimal frontline treatment strategy remains controversial [18]. Moreover, the results of DH are generally not readily available because of long turnaround times and high costs. We generally stratify the risks of high-grade B-cell lymphoma by clinical profiles, including tumor burden, DE status, R-IPI score, and Lugano staging in our real-world practice. However, the impact of DH requires further investigation. Herein, a retrospective study was conducted to analyze the prognostic impact of DH in DLBCL patients in our institution. This study further attempted to develop an algorithm to predict the DH genetics of DLBCL using clinical parameters.

## 2. Materials and Methods

### 2.1. Patients

The medical records of 126 consecutive DLBCL patients newly diagnosed between January 2018 and July 2020 at Taichung Veterans General Hospital with corresponding clinicopathological data were reviewed. Patients who did not receive FISH because of the inadequate sample size (*n* = 29), primary central nervous system involvement (*n* = 3), and no regular follow-up (*n* = 2) were excluded. Eventually, 92 patients were included in the current study. To investigate the clinical features and prognostic impact of DH genetics, the 92 patients were further stratified into the DH group (*n* = 14) and the non-DH group (*n* = 78). 

The bulky disease was established when a tumor diameter was ≥6 cm [19]. We measured the maximal standardized uptake values (SUVs) using the [(18)F]-fluoro-2-deoxy-d-glucose positron emission tomography (FDG-PET) with a whole-body acquisition from the groin to the head in 67 of the 92 patients. The general condition of the study cohort was assessed using the Charlson comorbidity index (CCI) score and Eastern Cooperative Oncology Group performance status (PS) [20]. In terms of the initial treatments, 85 patients received intent-to-cure treatments according to the physician’s choice. Treatment regimens were stratified into standard R-CHOP (*n* = 65), less intensive chemotherapy (R-COP) (*n* = 7), and more intensive chemotherapy (R-EPOCH or R-hyperCVAD) (*n* = 13). This study was approved by the Review Board of Taichung Veterans General Hospital (CE21362A) and was conducted following the Declaration of Helsinki. The institutional review board waived the requirement for patients’ informed consent because of the retrospective study design.

### 2.2. Immunohistochemistry Analysis

The c-Myc, Bcl-2, Bcl-6, CD10, and MUM1 expressions in DLBCL cells were determined by IHC staining. Briefly, the samples were prepared using 3 µm formalin-fixed paraffin-embedded (FFPE) tissue sections. The primary antibody probes utilized were c-myc (clone EP121; Cat# BSB-6580, BioSB, Santa Barbara, CA, USA), Bcl-2 (clone 124; Cat# M0887, DAKO, Santa Clara, CA, USA), Bcl-6 (clone RBT-bcl6; Cat# BSB-5082, BioSB, Santa Barbara, CA, USA), CD10 (clone SP67; Cat# 790-4506, VENTANA, Tucson, AZ, USA), and MUM1 (clone MRQ-43; Cat# 790-4529, VENTANA, Tucson, AZ, USA). We utilized the Ventana benchmark ULTRA IHC staining system for the IHC staining. We defined DE as when ≥40% and ≥50% of DLBCL cells were positive for c-myc and BCL-2, respectively [21]. The cell of origin determination was based on Hans’ algorithm [22]. 

### 2.3. Double Hit Analysis

For the DH analysis, we employed FISH to detect MYC, BCL2, or BCL6 translocation. The dual-color break-apart rearrangement probes: SureFISH IGH MYC DF P20 (8q24.21) (G111425), Santa Clara, CA, USA, SureFISH BCL2 P20(18q21.33) (G111421), Santa Clara, CA, USA, and SureFISH BCL6 BA P20(3q27.3) (G111422), Santa Clara, CA, USA, were employed. The probe signals for monolayers of ≥200 DLBCL cell nuclei were counted under a fluorescence microscope at ×100 magnification. Genetic alteration of DH was established to occur when the probe signals exhibited a ≥15% threshold relative to the number of nuclei.

### 2.4. Statistical Analysis

Continuous and categorical variables between the DH and non-DH groups were compared using the Mann–Whitney U test and the Chi-squared test or Fisher’s exact test, as indicated. Numerical data are presented as the median ± standard deviation. The OS was defined as the period from the first day of treatment to death by any cause or the study censored day (30 June 2021). We performed OS analysis restricted in patient received immunochemotherapy. Patients who underwent palliative care only were excluded (*n* = 2 in the DH group; *n* = 5 in the non-DH group). Univariate and multivariate Cox regression models were used to determine the prognostic relevance quantified as hazard ratios (HRs), with 95% confidence intervals (CIs). The Kaplan–Meier survival curve was applied to estimate the OS. Sensitivity and specificity tests were employed to evaluate the DH prediction accuracy. All the tests were analyzed using the Statistical Package for the Social Sciences (IBM SPSS version 22.0; International Business Machines Corp, New York, NY, USA). The statistical significance was set at *p* < 0.05.

## 3. Results

### 3.1. Comparison of Clinical and Biological Features between the DH and Non-DH Groups 

The sex, age, performance status, staging, R-IPI score, lactate dehydrogenase (LDH), uric acid, maximal SUV, and cell of origin were not significantly different between the DH and non-DH groups. However, the DH group had more occurrences of bulky diseases than the non-DH group (64.3% vs. 28.2%; *p* = 0.013). Moreover, more patients in the DH group were DE (+) (50.0% vs. 21.8%; *p* = 0.044). Concerning the treatments, the initial regimens were similar between groups (Table 1).

### 3.2. Outcome Comparison between the DH and Non-DH Groups

Survival analysis was performed on patients who received immunochemotherapies. The CR rates in the DH and non-DH groups were 58.3% and 64.4%, respectively (*p* = 0.687). With a median follow-up of 19.2 months (range, 1.5–40.9 months), the three-year OS rates of patients with and without DH were 33.3% and 52.2%, respectively (*p* = 0.016) (Figure 1). 

Regarding mortality, the two groups had similar causes of death (*p* = 0.663). Underlying lymphoma remained the leading cause of death in the two groups, accounting for 70.0% and 53.6%, respectively (Table 1).

### 3.3. DH Genetics as an Independent Factor Associated with Inferior OS

We employed Cox regression analysis to investigate the impact of DH genetics on OS in DLBCL patients receiving immunochemotherapies. The univariate analysis revealed that DH (HR: 2.63; 95% CI: 1.16–5.93; *p* = 0.020), age ≥ 65 years (HR: 2.74; 95% CI: 1.32–5.67; *p* = 0.007), a high CCI score (HR: 1.35; 95% CI: 1.13–1.60; *p* = 0.001), uric acid level (HR: 1.13; 95% CI: 1.02–1.25; *p* = 0.019), maximal SUV (HR: 1.09; 95% CI: 1.02–1.17; *p* = 0.016), advanced stage (HR: 6.58; 95% CI: 2.00–21.66; *p* = 0.002), bulky disease (HR: 2.17; 95% CI: 1.07–4.42; *p* = 0.032), and poor risk R-IPI score (HR: 3.21; 95% CI: 1.51–6.83; *p* = 0.002) were significantly associated with an inferior OS (Table 2). Multivariate analysis further validated some of the results, showing advanced stage (HR: 5.00; 95% CI: 1.36–18.35; *p* = 0.015), and CCI score (HR: 1.30; 95% CI: 1.03–1.63; *p* = 0.025) were independent factors for mortality. However, DH was not substantially associated with inferior OS in this multivariate analysis (HR: 1.62; 95% CI: 0.65–4.06; *p* = 0.300) (Table 2).

### 3.4. Clinical Data of DH (+) DLBCL Patients 

Since DH genetics could be factor for mortality in DLBCL patients, we analyzed the clinical data of 14 DH (+) DLBCL patients in the study cohort. The CR rate of DH (+) DLBCL after frontline therapy was 58.3% (7/12). The three-year OS rate was 33.3% among patients undergoing immunochemotherapies (4/12). All the DH (+) DLBCL patients that did not achieve CR died of the underlying lymphoma. Table 3 shows the causes of death.

### 3.5. Bulky Disease with DE Immunopositivity for Identifying DH DLBCL

Since DH was associated with a low CR rate and increased mortality in DLBCL patients, a rapid and reliable surrogate marker to predict DH in newly diagnosed DLBCL patients is essential. Our results showed that the sensitivities of DE and the bulky disease for DH prediction were 50.0% and 64.3%, while the specificities were 78.2% and 71.8%, respectively. DE appears more accurate than the bulky disease for predicting DH (73.9% vs. 70.7%). Furthermore, by combining the DE and bulky disease, a specificity of 89.7% for DH prediction was achieved (Table 4).

## 4. Discussion

In the current study, it was found that DH genetics constituted 15.2% of newly diagnosed DLBCL cases. Among DH DLBCL patients, only 50.0% were positive for DE immunophenotype. DH genetics, but not DE immunopositivity, was a factor for inferior OS in terms of survival. However, we established that the positive DE immunophenotype and bulky disease were both indicative of DH genetics. A positive DE immunophenotype and a bulky disease could predict DH genetics with specificities of 78.2% and 71.8%, respectively. Furthermore, a combination of bulky disease and positive DE immunophenotype could achieve a specificity of 89.7% for predicting DH genetics in newly diagnosed DLBCL patients. 

The impact of DH genetics on the OS of DLBCL is increasing [23]. Our study showed that DLBCL patients with DH genetics have a shorter OS by three years compared to those without DH genetics (33.3% vs. 52.2%; *p* = 0.016). The MYC gene alterations may be the primary cause of this. The MYC gene was initially detected in Burkitt lymphoma as a potent oncogene and is associated with aggressive clinical behavior [24]. The role of the MYC gene has been extensively investigated. Not only is it a transcription factor that causes cell proliferation, but also it plays a role in the induction of apoptosis [25]. Aggressive lymphomas appear to have acquired additional oncogenic alterations that cooperate with MYC dysregulation by counteracting its pro-apoptotic function [26]. Moreover, studies have suggested that MYC translocation arises in the germinal center (GC) microenvironment because of the role of massive proliferation and GC phenotype [27]. In the cell of origin model, DH genetics is more frequent in the GCB DLBCL group, while DE immunopositivity is more frequent in the non-GCB subset [28]. However, our study did not reveal the same results, presumably because of the limited number of patients.

Additionally, the current study analyzed the correlation between DH genetics and DE immunophenotype. A study by Scott et al. [14] showed that 75.0% of DLBCL patients with DH genetics had a positive DE immunophenotype; however, the reverse was not necessarily true. Our results showed that only 50.0% of DLBCL patients with DH genetics were positive for the DE immunophenotype, which was lower than that reported by Scott et al. More than existing DH genetics, chromosomal translocations, MYC gene amplification, and post-translational processes could be the potential mechanisms leading to increased MYC protein expression [29]. Moreover, various genetic backgrounds in different cohorts may also be one of the reasons for this difference. Further studies are required to validate this discrepancy. 

Regarding survival, the current study found that DH DLBCL patients had a lower OS rate than non-DH DLBCL patients. Interestingly, the CR rates between the two groups were not substantially different. A previous study revealed that DH DLBCL might benefit more from intensified chemotherapy [30]. Therefore, we examined the disease course of each DH DLBCL in our study cohort. We established that only two (cases 8 and 14) of the 14 patients received chemotherapeutic regimens that are more intense than R-CHOP as their frontline treatment. This reflects the fact that the early identification of DH DLBCL remains a challenge. Thus, the identification of potential DLBCLs using surrogate markers is crucial. 

DH genetics has been associated with elevated LDH, high-risk IPI scores, and advanced stages [7]. Our analyses showed that DLBCL patients with DH genetics were more likely to have the bulky disease. Although the prognostic value of the bulky disease appears to be less prominent in the era of rituximab [31], we attempted to utilize bulky disease and DE immunophenotype as surrogate markers to predict DH genetics in DLBCL. With a specificity of 89.7%, the findings suggested that DLBCL patients without the bulky disease and DE immunophenotype were less likely to harbor DH genetics. Thus, standard R-CHOP treatment may be appropriate for these patients. 

The small number of patients and the retrospective nature of the study were the primary limitations of the current research. Although our retrospective cohort showed that DH genetics were associated with bulky disease and an inferior OS in DLBCL patients, studies with more patients and prospective design are needed to validate our results. In summary, our study demonstrated that DH genetics, not DE immunophenotype, could be a factor for an inferior OS in DLBCL. The combination of a positive DE immunophenotype and bulky disease simultaneously could achieve a specificity of 89.7% for predicting DH genetics in newly diagnosed DLBCL patients. Intensified chemotherapeutic regimens or frontline autologous stem cell transplantation may be considered for fit DLBCL patients with these two features.

## Figures and Tables

**Figure 1 diagnostics-12-01106-f001:**
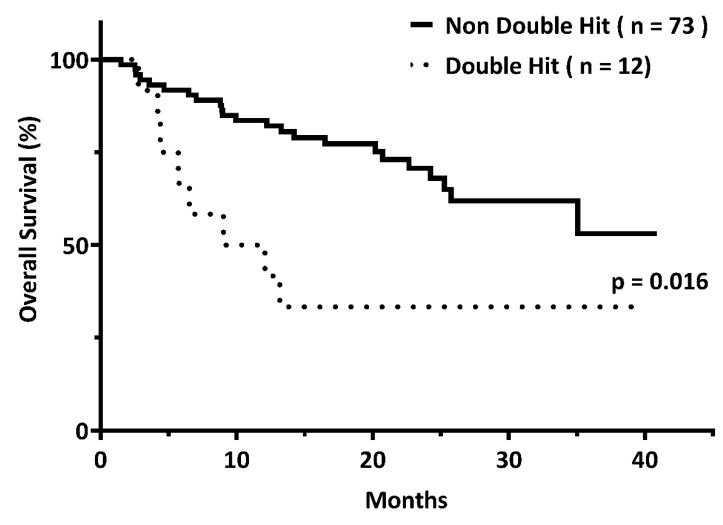
The three-year overall survival rates among patients with (*n* = 12) and without (*n* = 73) double hit (DH) genetics were 33.3% and 52.2%, respectively (*p* = 0.016).

**Table 1 diagnostics-12-01106-t001:** Patients’ characteristics and comparison of outcomes.

	All Patients (*n* = 92)	Double Hit (*n* = 14)	Non-Double Hit (*n* = 78)	*p*-Value
Gender, *n* (%)				0.909 ^a^
Male	48 (52.2%)	8 (57.1%)	40 (51.3%)	
Female	44 (47.8%)	6 (42.9%)	38 (48.7%)	
Age when diagnosed, *n* (%)				0.518 ^a^
<65	50 (54.3%)	6 (42.9%)	44 (56.4%)	
≥65	42 (45.7%)	8 (57.1%)	34 (43.6%)	
Performance status, *n* (%)				0.060 ^b^
≤2	81 (88.0%)	10 (71.4%)	71 (91.0%)	
>2	11 (12.0%)	4 (28.6%)	7 (9.0%)	
Stage, *n* (%)				0.134 ^b^
Limited (Stage 1–2)	30 (32.6%)	2 (14.3%)	28 (35.9%)	
Advanced (Stage 3–4)	62 (67.4%)	12 (85.7%)	50 (64.1%)	
Bulky disease, *n* (%)				0.013 ^b^
No	61 (66.3%)	5 (35.7%)	56 (71.8%)	
Yes	31 (33.7%)	9 (64.3%)	22 (28.2%)	
R-IPI score, *n* (%)				0.640 ^a^
Lower risk (0–2)	48 (52.2%)	6 (42.9%)	42 (53.8%)	
Higher risk (3–5)	44 (47.8%)	8 (57.1%)	36 (46.2%)	
LDH (U/L), median (range)	315 (111–4461)	360 (169–1760)	303 (111–4461)	0.277 ^c^
Uric acid (mg/dL), median (range)	5.5 (1.5–19.0)	5.2 (3.5–16.4)	5.6 (1.5–19.0)	0.721 ^c^
Maximal SUV (/1 h), median (range)	12.6 (2.5–27.4)	13.1 (4.1–25.2)	12.4 (2.5–27.4)	0.436 ^c^
CCI score, median (range)	4 (2–11)	5 (2–11)	4 (2–11)	0.659 ^c^
Double expresser, *n* (%)				0.044 ^b^
No	68 (73.9%)	7 (50.0%)	61 (78.2%)	
Yes	24 (26.1%)	7 (50.0%)	17 (21.8%)	
Cell of origin, *n* (%)				0.122 ^b^
GCB	30 (34.1%)	7 (53.8%)	23 (30.7%)	
Non-GCB	58 (65.9%)	6 (46.2%)	52 (69.3%)	
Treatment, *n* (%)				0.501 ^a^
R + intensified chemotherapy	13 (14.1%)	2 (14.3%)	11 (14.1%)	
R-CHOP	65 (70.7%)	8 (57.1%)	57 (73.1%)	
R-COP	7 (7.6%)	2 (14.3%)	5 (6.4%)	
Palliative care	7 (7.6%)	2 (14.3%)	5 (6.4%)	
Treatment response, *n* (%)				0.687 ^a^
Non-CR	31 (36.5%)	5 (41.7%)	26 (35.6%)	
CR	54 (63.5%)	7 (58.3%)	47 (64.4%)	
Mortality, *n* (%)				0.028 ^a^
Alive	54 (58.7%)	4 (28.6%)	50 (64.1%)	
Death	38 (41.3%)	10 (71.4%)	28 (35.9%)	
Causes of death, *n* (%)				0.663 ^a^
Disease related	22 (57.9%)	7 (70.0%)	15 (53.6%)	
Complication	11 (28.9%)	2 (20.0%)	9 (32.1%)	
Others	5 (13.2%)	1 (10.0%)	4 (14.3%)	

Data were compared using ^a^ Chi-square test, ^b^ Fisher’s Exact test, and ^c^ Mann–Whitney U test. LDH: lactate dehydrogenase; SUV: standardized uptake value; CCI: Charlson comorbidity index; GCB: germinal center B-cell; R: rituximab; CR: complete remission.

**Table 2 diagnostics-12-01106-t002:** Risk factors for diffuse large B cell lymphoma.

Clinical Variables	Univariate			Multivariate		
	HR	95% CI	*p*-Value	HR	95% CI	*p*-Value
Age, years						
<65	1.00			1.00		
≥65	2.74	1.32–5.67	0.007	1.46	0.57–3.71	0.432
Gender						
Male	1.00					
Female	0.78	0.38–1.59	0.489			
Performance status						
≤2	1.00					
>2	2.17	0.76–6.22	0.150			
Stage						
Limited (Stage: 1–2)	1.00			1.00		
Advanced (Stage: 3–4)	6.58	2.00–21.66	0.002	5.00	1.36–18.35	0.015
Bulky, *n* (%)						
No	1.00			1.00		
Yes	2.17	1.07–4.42	0.032	1.80	0.78–4.16	0.169
R-IPI score						
Non-poor risk (0–2)	1.00			1.00		
Poor risk (3–5)	3.21	1.51–6.83	0.002	1.21	0.52–2.85	0.655
CCI score	1.35	1.13–1.60	0.001	1.30	1.03–1.63	0.025
LDH (U/L)	1.00	1.00–1.00	0.197			
Uric acid (mg/dL)	1.13	1.02–1.25	0.019			
Maximal SUV (/1 h)	1.09	1.02–1.17	0.016			
Double hit						
No	1.00			1.00		
Yes	2.63	1.16–5.93	0.020	1.62	0.65–4.06	0.300
Double expresser						
No	1.00					
Yes	1.47	0.69–3.12	0.320			
Cell of origin						
GCB	1.00					
Non-GCB	1.11	0.51–2.42	0.800			

Data analysis by the Cox proportional hazard model. HR: hazard ratio; CI: confidence interval; CCI: Charlson comorbidity index; LDH: lactate dehydrogenase; SUV: standardized uptake value; GCB: germinal center B-cell.

**Table 3 diagnostics-12-01106-t003:** Detailed profile of double hit (+) diffuse large B cell lymphoma patients.

No.	Age	Stage	R-IPI	Bulky	DE	Frontline Treatment	CR	Alive	Cause of Death	PFS (Days)	OS (Days)
1	75	4	5	Yes	Yes	Palliative care	No	No	Disease related	14	14
2	64	1	1	No	No	R-CHOP × 6	Yes	Yes		1203	1203
3	75	2	2	Yes	No	R-COP × 8	No	No	Disease related	257	368
4	67	3	2	Yes	No	R-CHOP × 4	Yes	No	Hepatitis B flare-up	174	174
5	54	4	2	No	No	R-CHOP × 8	Yes	Yes		1176	1176
6	77	3	3	Yes	No	R-COP × 2	No	No	Disease related	85	85
7	54	4	2	No	Yes	(R-CHOP + IT) × 5	Yes	No	Septic shock	128	128
8	46	4	2	No	No	(R-CHOP + IT, R-MA) × 2	No	No	Disease related	223	275
9	80	4	4	Yes	Yes	Palliative care	No	No	Disease related	43	43
10	71	3	4	Yes	No	R-CEOP × 5	No	No	Disease related	158	401
11	48	4	3	Yes	Yes	R-CHOP × 1	No	No	Disease related	58	134
12	70	4	3	Yes	Yes	R-CHOP × 6	Yes	Yes		866	866
13	66	4	3	No	Yes	R-CHOP × 6	Yes	No	Aortic stenosis with heart failure	199	199
14	41	3	3	Yes	Yes	R-EPOCH × 6	Yes	Yes		1157	1157

R-IPI: revised International Prognostic Index; DE: double expresser; CR: complete remission; PFS: progression-free survival; OS: overall survival; IT: intrathecal chemotherapy; R-CHOP: rituximab plus cyclophosphamide, doxorubicin, vincristine, prednisone; R-COP: rituximab, cyclophosphamide, vincristine, prednisone; R-EPOCH: rituximab, etoposide, prednisone, vincristine, cyclophosphamide, doxorubicin; R-MA: rituximab, methotrexate, cytarabine.

**Table 4 diagnostics-12-01106-t004:** Prediction of double hit genetics.

		Double Hit			Sensitivity (%)	Specificity (%)	PPV (%)	NPV (%)	Accuracy (%)
		Total	Yes	No					
Double expresser	Yes	24	7	17	50.0	78.2	29.2	89.7	73.9
	No	68	7	61					
Bulky disease	Yes	31	9	22	64.3	71.8	29.0	91.8	70.7
	No	61	5	56					
Double expresser + bulky disease	Yes	13	5	8	35.7	89.7	38.5	88.6	81.5
	No	79	9	70					

PPV: positive predictive value; NPV: negative predictive value.

## Data Availability

Not applicable.

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
