# Peer review of "Clinical Features and Immunophenotypes of Double-Hit Diffuse Large B-Cell Lymphoma"

_diagnostics, 2022, doi:10.3390/diagnostics12051106_

Round 1

Reviewer 1 Report

The authors presented a revised version of the previous manuscript addressing the request of reviewer 2. Since the topic is interesting, but the sample size is limited, I will suggest again to reduce it to a short report.

Author Response

Thanks for your comments. 

Reviewer 2 Report

This version is improved comared to the previous version, this can be acceptable for publication.

Author Response

Thanks for your comments. 

Round 2

Reviewer 1 Report

No further comment

This manuscript is a resubmission of an earlier submission. The following is a list of the peer review reports and author responses from that submission.

Round 1

Reviewer 1 Report

The manuscript is clear, and is presented in a well-structured manner, however I will suggest to reduce it to a brief report, since it confirms data already known. The novelty is related to the suggestion of 'surrogate markers' of impaired outcome (combination of positive DE immunophenotype and bulky disease) that could help clinicians and sites who do not have cytogenetic facilities to help clinical decisions. 

I will focus mostly on the suggestion 

  • General concept comments

    Article: highlighting areas of weakness, the testability of the hypothesis, methodological inaccuracies, missing controls, etc.

    Review: commenting on the completeness of the review topic covered, the relevance of the review topic, the gap in knowledge identified, the appropriateness of references, etc.

    These comments are focused on the scientific content of the ma

    nuscript and should be specific enough for the authors to be able to respond.

  • Spe
  • cific comments referring to line numbers, tables or figures that point out inaccuracies within the text or sentences that are unclear. These comments should also focus on the scientific content and not on spelling, formatting or English language problems, as these can be addressed at a later stage by our internal staff.

General questions to help guide your review report for research articles

  • Are the cited references current (mostly within the last 5 years)? Does it include an abnormal number of self-citations?
  • Is the manuscript scientifically sound and is the experimental design appropriate to test the hypothesis?
  • Are the manuscript’s results reproducible based on the details given in the methods section?
  • Are the figures/tables/images/schemes appropriate? Do they properly show the data? Are they easy to interpret and understand? Are the data interpreted appropriately and consistently throughout the manuscript? Please include details regarding the statistical analysis or data acquired from specific databases.

Reviewer 2 Report

this is a retrospective study of a limited number of DLBCL with 14 DH DLBCL which is the core of the manuscript. Some limitations are observed:

1- out of these 14 cases, not all patients recieved adapted therapies (2 palliative treatments and 2 R-COP which are not adapted to the disease). OS analysis is not easy to understand ans the same description of the non DH DLBCL should be presented, since 2 of the DH patients died of unrelated diseases.

2-GCB subtype analysis could also be presented (COO assessed  by IHC of other technic?)